# Autosomal Dominant Hypocalcemia Type 1 and Neonatal Focal Seizures

**DOI:** 10.3390/children10061011

**Published:** 2023-06-03

**Authors:** Raluca Ioana Teleanu, Marlene Alexandra Sarman, Diana Anamaria Epure, Margarita Matei, Ioana Roşca, Eugenia Roza

**Affiliations:** 1Faculty of Medicine, Clinical Neurosciences Department, Peadiatric Neurology, “Carol Davila” University of Medicine and Pharmacy, 050474 Bucharest, Romania; raluca.teleanu@umfcd.ro (R.I.T.); ioanarosca76@yahoo.com (I.R.); eugenia.roza@umfcd.ro (E.R.); 2Pediatric Neurology Department—“Dr. Victor Gomoiu” Children’s Hospital, 022102 Bucharest, Romania; epurediana@gmail.com; 3Endocrinology Department—“Dr. Victor Gomoiu” Children’s Hospital, 022102 Bucharest, Romania; mateimargarita@yahoo.com; 4“Prof. Panait Sarbu” Clinical Hospital, 060251 Bucharest, Romania

**Keywords:** neonatal hypocalcemia, seizures, hypoparathyroidism, hypercalciuria, calcium-sensing receptor

## Abstract

Autosomal dominant hypocalcemia type 1 (ADH1) is a rare form of hypoparathyroidism that is characterized by gain-of-function mutations in the *CASR* gene, which provides instructions for producing the protein called calcium-sensing receptor (CaSR). Hypocalcemia in the neonatal period has a wide differential diagnosis. We present the case of a female newborn with genetic hypoparathyroidism (L125P mutation of CASR gene), hypocalcemia, and neonatal seizures due to the potential correlation between refractory neonatal seizures and ADH1. Neonatal seizures were previously described in patients with ADH1 but not in association with the L125P mutation of the CASR gene. Prompt diagnosis and management by a multidisciplinary and an appropriate therapeutic approach can prevent neurological and renal complications.

## 1. Introduction

Hypocalcemia can be a life-threatening condition with symptoms ranging from mild symptoms to cardiac dysrhythmias, tetany, seizures, or laryngospasm in severe acute conditions. Hypocalcemia often develops as a consequence of vitamin D deficiency, hypoparathyroidism, or renal disease and occasionally can be part of a genetic syndrome [1,2]. Autosomal dominant hypocalcemia type 1 (ADH1) is a rare form of hypoparathyroidism caused by gain-of-function (GoF) mutations in the CASR gene, which increases the sensitivity of the protein called calcium-sensing receptor (CaSR) to extracellular ionized calcium, leading to an early stimulation of the G11 (a signaling protein) (Figure 1). This protein is highly expressed in the parathyroid glands and kidneys. Consequently, both PTH secretion and renal reabsorption of calcium are suppressed at normal serum calcium levels. Activating mutation of the alpha subunit of the G11 protein (GNA11) leads to autosomal dominant hypocalcemia type 2 (ADH2) [3].

Although most cases of isolated hypoparathyroidism are sporadic, some exhibit X-linked, autosomal recessive, or autosomal dominant inheritance. Usually, patients present with hypocalcemia, hypomagnesemia, hyperphosphatemia with low-normal levels of PTH, and hypercalciuria [3,4]. In ADH1, hypercalciuria presents a particularly challenging issue, arising from two distinct mechanisms that operate independently. Initially, reduced levels of PTH, which typically stimulate calcium reabsorption from the primary filtrate, lead to relative hypercalciuria. Subsequently, the heightened activation of the mutated CASR by extracellular calcium in the distal renal tubules causes further pronounced hypercalciuria [3]. While most impacted individuals may not exhibit any symptoms, prevalent manifestations of this condition include carpopedal spasms, paresthesia, and neuromuscular irritability [5]. In comparison, ADH2 patients may exhibit a slightly milder phenotype concerning hypocalcemia compared to those with ADH1. Furthermore, urinary calcium elevations in individuals with ADH2 seem less marked [3]. Patients with severe hypocalcemia can develop seizures—the most common presentation [6], usually in infancy or childhood [7,8,9,10,11]. The electroencephalogram in hypocalcemic seizures may present focal spikes and rhythmic, high-voltage discharges over the frontocentral region that rapidly generalize [12]. These EEG abnormalities emerge as a result of neuronal hyperexcitability and can be entirely reversible following the amelioration of serum calcium concentration [13]. A study conducted by R. Vargas et al. declared the L125P mutant is the most potent GoF mutation, and it is associated with a Bartter-like syndrome (renal sodium chloride depletion that is accompanied by secondary hyperaldosteronism and hypokalemia) [14].

It is important to differentiate hypoparathyroidism due to activating mutations of the calcium-sensing receptor gene from other etiologies in order to allow proper management of electrolyte abnormalities. The key feature of ADH1 is the association of hypocalcemia, hypercalciuria, and decreased or low-normal PTH levels [3].

## 2. Case Report

A 28-day-old girl was referred to our clinic to investigate recurrent focal to bilateral tonic–clonic seizures. The parents noted abnormal movements since birth, initially 1–3 episodes/day that increased in frequency and duration (4–5 seizures/hour lasting between 1–2 min) one day prior to the admission to our clinic. The girl was born following a full-term, uneventful pregnancy. The mother was infected with SARS-CoV-2 during the third trimester. There was no family history of epilepsy, neurological, or other metabolic disorders, including hypocalcemia. On admission, long-term video-electroencephalography (v-EEG) was performed. The ictal recording showed right focal to bilateral clonic seizures (right hemifacial clonic movements spread to the superior and inferior right limbs, with opisthotonic posture at the end of the seizure) (Figure 2). Interictal v-EEG showed normal background activity with rare spike and wave discharges in the left frontocentral regions of the brain. 

Routine blood tests revealed hypocalcemia, total serum calcium 5.94 mg/dL (9.0–11.0 mg/dL); hypomagnesemia, 1.41 mg/dL (1.70–2.56/mg/dL); and hyperphosphatemia, 10.48 mg/dL (3.7–6.5/mg/dL) with normal liver and kidney function and negative inflammation markers. Hypercalciuria (milky urine) and hypophosphaturia were present. The parathyroid hormone levels were low, 2.3 pg/mL (15–65 pg/mL). The serum Na, K, and Cl levels were 137 mmol/L (135–148 mmol/L), 5.22 (3.7–5.3 mmol/L), and 104.4 mmol/L (93–112 mmol/L), respectively.

The main challenges in managing our patient consisted of controlling the seizures and rapidly correcting life-threatening hypocalcemia. Given the substantial probability of ADH1 at the time of diagnosis, the correlation between this pathology and epilepsy [11,15], and the seizure frequency and severity, we prioritized seizure cessation by means of antiepileptic medication. A loading dose of phenobarbital (20 mg/kg) was administered, followed by oral administration, but the seizures persisted even with the subsequent addition of intravenous levetiracetam (50 mg/kg/day). Upon adding topiramate (3 mg/kg/zi) to the treatment regimen, the seizures stopped. Considering that after initiation of phenobarbital and then levetiracetam the neonate experienced drowsiness and poor feeding, we opted against increasing the levetiracetam to the maximal dosage of 60 mg/kg/day and instead introduced a third anticonvulsant to the regimen with gradual discontinuation of the prior antiepileptic medications. In our case, the antiepileptic treatment with topiramate led to seizure remission at a decreased serum calcium level despite the simultaneous administration of intravenous calcium gluconate and oral calcitriol. The treatment strategy implemented for rectifying severe hypocalcemia (calcium: 5.94 mg/dL) in our patient involved the following steps: first, as the electrocardiogram (ECG) showed a prolonged QT interval secondary to hypocalcemia, an intravenous bolus of gluconic calcium (2 mL/kg) diluted in 5% dextrose solution was urgently administered over a 10-min span. Immediately following this, gluconic calcium was administered at a dosage of 50 mg of elemental calcium per kilogram per day (5.5 mL/kg/day) along with a vitamin D analogue (calcitriol) at a dosage of 0.06 µg per kilogram per day (0.02–0.06 µg/kg/day in two equally divided doses). Before admission to the hospital, the patient received oral cholecalciferol 500 UI/day for rickets prophylaxis. However, at the time of the diagnosis of hypoparathyroidism, it was decided to initiate calcitriol and cease the administration of cholecalciferol, considering that the bioconversion of this vitamin requires the parathyroid hormone. Intravenous administration of calcium was switched to an oral route when the symptoms were managed, and the calcium level surpassed the threshold of 7.5 mg/dl [16] with a daily dose of 2.2 mL/kg/day. We titrated the doses of oral calcitriol up to 0.08 ug/kg/day. Serum magnesium, phosphate, alkaline phosphatase, and calcium levels normalized, and the calcium–phosphorus (Ca × P) product was reduced (Table 1). Adequate control of the calcium and phosphate homeostasis was reached by the tenth day of hospitalization. The goal was to maintain the calcium–phosphorus product below 65 mg2/dL2 and urinary calcium excretion < 4 mg/kg/day [1,17]. A thiazide diuretic was considered as a hypercalciuria control, but as the renal and bladder ultrasonography and the nephrological evaluation were normal, it was postponed. In the event of failure to correct calciuria, a contingency plan was set to administer a thiazide diuretic at a dosage of 0.5 mg/kg/day based on the advice of the nephrologist. The 1.5 T cerebral magnetic resonance imaging (MRI) was lesion-negative. Ophthalmological examination showed no abnormalities.

Considering the association of neonatal hypocalcemia with hypoparathyroidism and seizures, DiGeorge Syndrome as well as other genetic disorders were taken into account. Multiplex ligation-dependent probe amplification (MLPA) along with whole exome sequencing (WES) testing were performed. The MLPA tested negative for 22q11 deletion (DiGeorge Syndrome), while the WES testing identified a de novo heterozygous calcium-sensing receptor (*CASR*) gene mutation with a c.3747 > C p (Leu125Pro) variant. The result was consistent with the genetic diagnosis of autosomal dominant hypocalcemia type 1 (Figure 3).

Over the next 5-month follow-up period, serum calcium levels were monitored monthly, and calcitriol doses were increased according to the physiological weight gain of the infant. Considering the mixed etiology of the seizures, i.e., a metabolic disorder with a genetic cause, the antiseizure medication was slowly withdrawn. The infant developed normally but experienced two short febrile seizures (~2 min in duration) at 7 and 10 months old despite having normal serum calcium levels. The first episode was a simple febrile seizure. In contrast, the second episode consisted of a focal motor seizure (clonic movements of the superior and inferior right limbs). Considering the time span between seizures and the seldom occurrence of spike and wave discharges on EEG, we decided not to resume the administration of antiepileptic medication. 

## 3. Discussion 

Various etiologies can contribute to neonatal seizures, requiring prompt identification and management of the seizure and its underlying cause. In the neonatal period, seizures represent the most prevalent neurological emergency, and unlike those occurring in infancy and childhood, they are frequently provoked by an acute cause [18]. Consequently, upon admission, primary factors associated with neonatal seizures were investigated, encompassing hypoxic-ischemic encephalopathy, infectious origins, metabolic disturbances, and genetic and structural anomalies. Our methodology entailed birth history, physical examination, laboratory examinations to evaluate potential electrolyte imbalances, hypoglycemia, hypocalcemia, video-electroencephalography, and neuroimaging techniques to identify the presence of hemorrhage, infarction, or cerebral structural anomalies. However, the severe hypocalcemia, hypoparathyroidism, and seizures in a neonate prompted further testing and revealed a genetic etiology explaining the clinical and paraclinical picture.

Our patient presented a rare case of severe hypocalcemia, a de novo pathogenic variant of the CASR gene (L125P). The same mutation was previously described in relation to a Bartter-like syndrome in a study conducted by R. Vargas et al. in a male patient with severe hypocalcemia associated with renal loss of NaCl, resulting in secondary hyperaldosteronism and hypokalemia. We reported this case to highlight a new possible phenotype–genotype association in neonatal epilepsy. Neonatal seizures were previously described in patients with ADH1 [19,20] but not in association with the L125P mutation of CASR gene.

A previous case report of *CASR* mutation (Phe788Cys) in a Japanese family described some members of the family experiencing seizures while others did not despite the severely low calcium level, indicating that seizures can occur independent of serum calcium levels in these patients [20]. An identical *CASR* mutation (Phe788Cys) was documented in a female patient who experienced upper limb myoclonic jerks starting at 18 years old. Computed tomography scans revealed basal ganglia calcifications [7]. A recent systematic review of ADH1 discovered 113 different *CASR* variants with a gain-of-function mutation in 338 patients, the most common variant being P221L, with no genotype–phenotype correlation. Conversely, the A843E variant is consistently linked to a more severe clinical presentation (all patients presented symptoms and electrolyte abnormalities). Additionally, just a single instance of a homozygous mutation in ADH1 has been documented, with the patient exhibiting mild symptoms and moderately low calcium levels [6].

In addressing the management of hypocalcemia in patients diagnosed with autosomal dominant hypocalcemia type 1, it is crucial to take into account that asymptomatic individuals should not be subjected to attempts of rectifying serum calcium concentrations. Conversely, for symptomatic patients, maintaining an optimal level of calcium (at the lower threshold of the normal range) and phosphatemia is advisable. The administration of calcium gluconate and vitamin D derivatives may result in an elevation of serum calcium levels. However, due to the activating mutation in the CASR gene inherent to ADH1, this augmentation could lead to an increase in urinary calcium excretion. This phenomenon carries associated risks, including renal lithiasis, nephrocalcinosis, and potential renal failure. Other unfavorable outcomes include calcifications within the basal ganglia and the development of cataracts. In contrast, thiazide diuretics prevent urinary calcium excretion and maintain a normal serum calcium concentration. Consequently, thiazide diuretics should be considered as the first-choice treatment for patients with asymptomatic hypocalcemia and hypercalciuria [21]. 

Human recombinant parathyroid hormone (rhPTH) represents a potential successful treatment of hypoparathyroidism in pediatric patients as it has demonstrated positive impacts on phosphatemia, Ca–P product, and hypercalciuria [22,23,24,25,26], but further studies are required [23] as currently no consensus on ADH1 treatment exists. However, this treatment carries a theoretical risk of osteosarcoma, which remains unconfirmed in human studies [27]. 

Another novel therapy in trial to treat hypoparathyroidism is the calcilytic encaleret (CLTX-305), a CaSR antagonist, that decreases the sensitivity of calcium-sensing receptors [28].

## 4. Conclusions

ADH1 is characterized by phenotypic heterogeneity. The association of hypocalcemia, hypercalciuria, and decreased PTH levels, which represents the core element of ADH1, led to the genetic testing in our case. To the best of our knowledge, this is the first case of neonatal seizures associated with the L125P pathogenic variant of CASR gene. In our case, multidisciplinary management was paramount for devising the correct approach. Further challenges are to be expected with respect to maintaining calcium levels within a normal range in order to avoid further complications, such as basal ganglia calcifications and nephrocalcinosis, as well as a risk of seizure recurrence given that our patient presented seizures independent of serum calcium levels. Genetic testing should always be considered in patients with “idiopathic hypoparathyroidism”, especially in symptomatic cases associated with neonatal seizures.

## Figures and Tables

**Figure 1 children-10-01011-f001:**
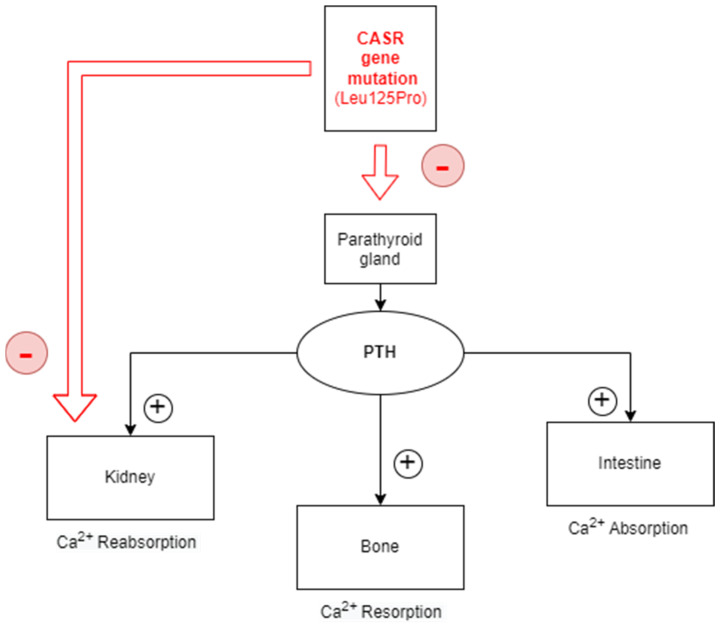
Calcium−sensing receptor (CASR) gene mutations affect calcium homeostasis. CASR gene provides instructions for producing the protein called calcium−sensing receptor (CaSR). The gain-of-function mutations in this gene lead to a receptor more sensitive to calcium levels in the blood and block the pathways that increase the serum calcium concentration (parathyroid glands and kidneys). Red box = inhibitory effect; CASR = calcium−sensing receptor; PTH = parathyroid hormone.

**Figure 2 children-10-01011-f002:**
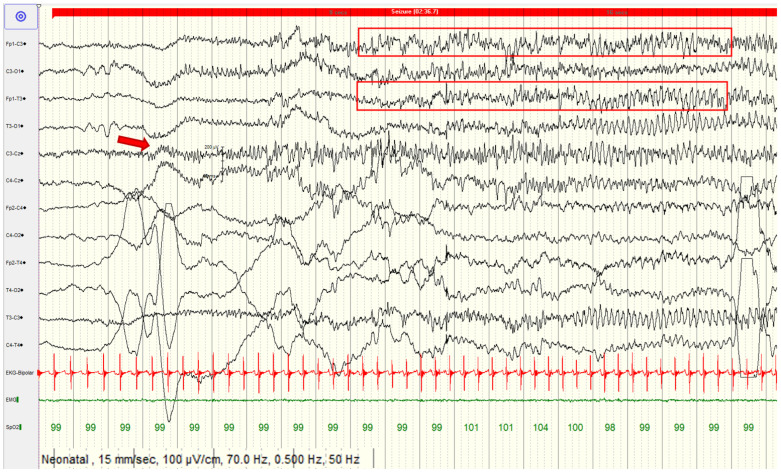
The long-term video-electroencephalographic monitoring (v-EEG-LTM) showing focal onset of the seizure in the left central leads (red arrow), spreading to the frontal areas (red box). Electrode Placement according to the International 10–20 System. Letters correspond to lobes—F (rontal), T (emporal), P (arietal), and O (ccipital). C stands for Central. Odd numbers underscript correspond to the left hemisphere, even numbers to the right hemisphere EKG = electrocardiogram; EMG = electromyography; SpO_2_ = peripheral capillary oxygen saturation.

**Figure 3 children-10-01011-f003:**
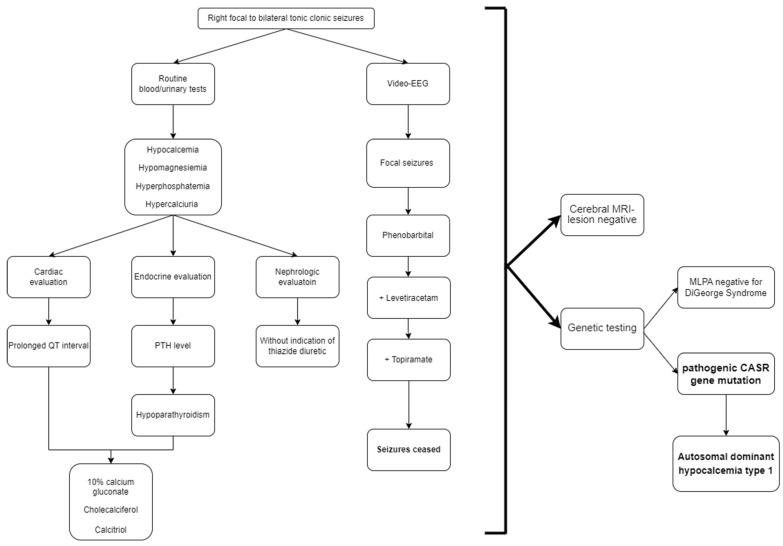
Diagnostic and management workflow of the patient. CASR = calcium-sensing receptor gene.QT interval = the time from the beginning of the QRS complex to the end of the T wave; PTH = parathyroid hormone; EEG= electroencephalogram; + = addition; MRI = Magnetic Resonance Imaging; MLPA = Multiplex Ligation-dependent Probe Amplification.

**Table 1 children-10-01011-t001:** Evolution of serum electrolyte levels.

Serum Electrolytes	At Admission	7 Days after Treatment Initiation	10 Days after Treatment Initiation
Calcium	5.94 mg/dL	7.94 mg/dL	9.69 mg/dL
Magnesium	1.41 mg/dL	1.56 mg/dL	1.72 mg/dL
Phosphorus	10.76 mg/dL	6.93 mg/dL	6.37 mg/dL

## Data Availability

No new data were created or analyzed in this study. Data sharing is not applicable to this article.

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
