# Peer review of "Autosomal Dominant Hypocalcemia Type 1 and Neonatal Focal Seizures"

_children, 2023, doi:10.3390/children10061011_

Round 1

Reviewer 1 Report

The authors present a a case study about:

 Autosomal dominant hypocalcemia type 1 and neonatal focal 2 seizures

Introduction:

Well written no comments

Case:

Figure 2 is of poor quality, resolution has to be increased and values of voltage has to be added.

Figure 1 and 3 are very clear and might be helpful for clinical decision making.

adequate

Reviewer 2 Report

1. Please clarify why do you want to report this case.  Refractory seizures is not enough as this has been previously described.  

Wu Y, Zhang C, Huang X, Cao L, Liu S, Zhong P. Autosomal dominant hypocalcemia with a novel CASR mutation: a case study and literature review. J Int Med Res. 2022 Jul;50(7):3000605221110489. doi: 10.1177/03000605221110489. PMID: 35818129; PMCID: PMC9280832

2. Separate discussion from conclusions

3. In the discussion address why this patient was treated with AED's along with calcium

4. Why patient Levetiracetam was only used yo 50 mg/Kg/d 

5. Why do you think topiramate worked on such small dose or by them was the calcium corrected?

6. Why do you think EEG is abnormal in hypocalcemia

7. Were other causes for neonatal seizures investigated?

8. Do you have a suggestion in which treatment algorythm should be done?

9. Were calcium alone should have been sufficient to treat the seizures

English is good.

Round 2

Reviewer 2 Report

Abstract

Line 13 Delete parenthesis

Line 14-16  This is not necessarily true, it could be actually the opposite.  One case does not allow to make this conclusion, should not get into the first part of the abstract.  This could be addressed in the discussion but you should present arguments.

Line 19 Why do you think is potential correlation, and what is really you are reporting the case, you need to decide one of them not many

Introduction

Line 41 autosomal in lower case

Figure 1 Need to decide reabsorption vs resorption. Keep consistency

Line 44 Needs to be adjusted to the left and the whole paragraph adequately formatted

Line 69-73 You should talk about other cases under discussion

The whole introduction should be shortened for instance line 82-87 could be summarized in one sentence.  Since the case is about the clinical presentation there is no reason to talk about treatment in great length. Focus in the clinical presentation, why do you want to report this case and the importance of this.

Line 112 delete pediatric neurology department

Line 114 no need for less frequent, just state the frequency, avoid narrative. How much did seizure frequency increased?

Are you sure, no endocrinological history, if not present this should be mentioned, relevant negative

Line 121 Fix the two lines prior to fig.2

Please add arrows on fig 2 to increase understanding. No everyone knows how to read EEGs

Please clarify treatment.  You mentioned first treatment was given in 10 minutes, but then you said it was increased.  Another bolus, was interrupted for how long.  I think one of the values of the paper is to share in as much detail as possible how did you actually treat.  At what point you introduce cholecalciferol and calcitriol

What was the rationale to switch from IV to oral.  Please clarify line 141-143

How old was the patient when he developed febrile seizures? What type, length

Do not separate initiation in the table

Avoid describing details about the case in the discussion, that is under case report

Discussion: Compare to similar articles, underly the importance of your case

 Conclusion should only contain what the facts of this case allow.

English is good.

Author Response

Abstract

Point 1. Line 13 Delete parenthesis

Response 1. We have modified it accordingly to your suggestion.

Point 2. Line 14-16  This is not necessarily true, it could be actually the opposite.  One case does not allow to make this conclusion, should not get into the first part of the abstract.  This could be addressed in the discussion but you should present arguments.

Response 2. Thank you. We decided to remove this paragraph.
Point 3. Line 19 Why do you think is potential correlation, and what is really you are reporting the case, you need to decide one of them not many

Response 3. We think that ADH1 and epilepsy are correlated as previous case reports exist in the literature describing this association. Therefore we have added the following:

  We present the case of a female newborn with genetic hypoparathyroidism (L125P mutation of CASR gene), hypocalcemia, and neonatal seizures due to the potential correlation between refractory neonatal seizures and ADH1. Neonatal seizures were previously described in patients with ADH1 but not in association with the L125P mutation of the CASR gene.

(19)         Nakajima, K.; Yamazaki, K.; Kimura, H.; Takano, K.; Miyoshi, H.; Sato, K. Novel Gain of Function Mutations of the Calcium-Sensing Receptor in Two Patients with PTH-Deficient Hypocalcemia. Intern. Med. Tokyo Jpn. 2009, 48 (22), 1951–1956. https://doi.org/10.2169/internalmedicine.48.2459.

(20)         Watanabe, T.; Bai, M.; Lane, C. R.; Matsumoto, S.; Minamitani, K.; Minagawa, M.; Niimi, H.; Brown, E. M.; Yasuda, T. Familial Hypoparathyroidism: Identification of a Novel Gain of Function Mutation in Transmembrane Domain 5 of the Calcium-Sensing Receptor1. J. Clin. Endocrinol. Metab. 1998, 83 (7), 2497–2502. https://doi.org/10.1210/jcem.83.7.4920.

Introduction

Point 4. Line 41 autosomal in lower case

Response 4. done
Point 5. Figure 1 Need to decide reabsorption vs resorption. Keep consistency

Response 5. Thanks for your valuable comments and remarks.We used the different terms as in the literature are described as two different mechanisms.

“When discussing the role of PTH on calcium homeostasis, the main effect in bone is to release calcium ions into the extracellular compartment to maintain normal ionized calcium concentrations. Osteoclasts are the cells responsible for bone resorption and release of minerals.”

(The Hormonal Regulation of Calcium Metabolism

Peter J. Tebben, Rajiv Kumar, in Seldin and Giebisch's The Kidney (Fourth Edition), 2008)

Renal Ca2+ reabsorption is essential for maintaining systemic Ca2+ homeostasis and is tightly regulated through the parathyroid hormone (PTH)/PTHrP receptor (PTH1R) signaling pathway. 

(Parathyroid hormone controls paracellular Ca2+ transport in the thick ascending limb by regulating the tight-junction protein Claudin14

Tadatoshi Sato, Marie Courbebaisse,  Proceedings of the National Academy of Sciences)

Point 6. Line 44 Needs to be adjusted to the left and the whole paragraph adequately formatted

Response 6. We have modified it accordingly to your suggestion

Point 7. Line 69-73 You should talk about other cases under discussion

The whole introduction should be shortened for instance line 82-87 could be summarized in one sentence.  Since the case is about the clinical presentation there is no reason to talk about treatment in great length. Focus in the clinical presentation, why do you want to report this case and the importance of this.

Response 7. Thank you for your suggestion. We have switched the paragraphs between the introduction, case report, and discussion accordingly.

Point 8. Line 112 delete pediatric neurology department

Response 8. We have changed it to “clinic”

A 28-day-old girl was referred to our clinic to investigate recurrent focal to bilateral tonic-clonic seizures.

Point 9. Line 114 no need for less frequent, just state the frequency, avoid narrative. How much did seizure frequency increase?

Response 9. We have added the following paragraph:

The parents noted abnormal movements since birth, initially 1-3 episodes/day that increased in frequency and duration (4-5 seizures/hour lasting between 1-2 minutes) one day prior to the admission to our clinic

Point 10. Are you sure, no endocrinological history, if not present this should be mentioned, relevant negative

Response 10. There was no endocrinological history. We have underlined this with the following sentence:

There was no family history of epilepsy, neurological or other metabolic disorders, including hypocalcemia
Point 11. Line 121 Fix the two lines prior to fig.2

Response 11. We have changed accordingly.

Point 12. Please add arrows on fig 2 to increase understanding. No everyone knows how to read EEGs
Respond 12. We have changed accordingly.

Figure 2. The long-term video-electroencephalographic monitoring (v-EEG-LTM) showing focal onset of the seizure in the left central leads (red arrow), spreading to the frontal areas (red box).

Point 13. Please clarify treatment.  You mentioned first treatment was given in 10 minutes, but then you said it was increased.  Another bolus, was interrupted for how long.  I think one of the values of the paper is to share in as much detail as possible how did you actually treat.  At what point you introduce cholecalciferol and calcitriol

What was the rationale to switch from IV to oral.  Please clarify line 141-143

Response 13. We have added the following paragraph for clarification:

First, an intravenous bolus of gluconic calcium (2 ml/kg) diluted in 5% dextrose solution was administered over a 10-minutes span with correction of the QT interval prolongation. Following this, gluconic calcium was administered at a dosage of 50 mg of elemental calcium per kilogram per day, along with a vitamin D analogue (calcitriol) at a dosage of 0.08 µg per kilogram per day.

Before admission to the hospital, the patient received cholecalciferol for rickets prophylaxis. However, at the time of the diagnosis of hypoparathyroidism, it was decided to initiate calcitriol and cease the administration of cholecalciferol, considering that the bioconversion of this vitamin requires the hypoparathyroid hormone.

Intravenous administration of calcium was switched to an oral route when the symptoms were managed and the calcium level surpassed the threshold of 7.5 mg/dl

Point 14. How old was the patient when he developed febrile seizures? What type, length

Response 14. We have added the following paragraph for clarification:

     The infant developed normally but experienced two short febrile seizures (~ 2 minutes in duration), at 7 and 10 months old, despite having normal serum calcium levels. The first episode was a simple febrile seizure. In contrast, the second episode consisted of a focal motor seizure (clonic movements of the superior and inferior right limbs)

Point 15. Do not separate initiation in the table

Response 15. We have changed accordingly.Point 16. Avoid describing details about the case in the discussion, that is under case report

Discussion: Compare to similar articles, underly the importance of your case

Response 16. We have changed accordingly.
 Point 17. Conclusion should only contain what the facts of this case allow.

Response 17. We have modified as follows:

     ADH1 is characterized by phenotypic heterogeneity. The association of hypocalcemia, hypercalciuria, and decreased PTH levels which represents the core element of ADH1, led to the genetic testing in our case. To the best of our knowledge, this is the first case of neonatal seizures associated with the L125P pathogenic variant of CASR gene. In our case, multidisciplinary management was paramount for a correct approach. Further challenges are to be expected with maintaining calcium levels within a normal range in order to avoid further complications like basal ganglia calcifications and nephrocalcinosis, as well as a risk of seizure recurrence given that our patient presented seizures independently of serum calcium levels. Genetic testing should always be considered in patients with “idiopathic hypoparathyroidism”, especially in symptomatic cases associated with neonatal seizures.

Round 3

Reviewer 2 Report

Only one more question,    Following this, (HOW LONG AFTER THE BOLUS, EXACLTY) gluconic calcium was administered at a dosage of 50 mg of elemental calcium per kilogram per day, along with a vitamin D analogue (calcitriol) at a dosage of 0.08 µg per kilogram per day.

Only one more question,    Following this, (HOW LONG AFTER THE BOLUS, EXACLTY) gluconic calcium was administered at a dosage of 50 mg of elemental calcium per kilogram per day, along with a vitamin D analogue (calcitriol) at a dosage of 0.08 µg per kilogram per day.
